# Progressive Local and Non-Local Interactive Networks with Deeply Discriminative Training for Image Deraining

Cong Wang
Shenzhen Campus of Sun Yat-sen University & The Hong Kong Polytechnic University
Shenzhen & Hong Kong, China
supercong94@gmail.com

Liyan Wang
Dalian University of Technology
Dalian, China
wangliyan@dlut.edu.cn

Jie Mu
Dongbei University of Finance and Economics
Dalian, China
jiemu@dufe.edu.cn

Chengjin Yu
Anhui University
Hefei, China
23073@ahu.edu.cn

Wei Wang*
Shenzhen Campus of Sun Yat-sen University
Shenzhen, China
wangwei29@mail.sysu.edu.cn

## Abstract

In this paper, we develop a progressive local and non-local interactive network with multi-scale cross-content deeply discriminative learning to solve image deraining. The proposed model contains two key techniques: 1) Progressive Local and Non-Local Interactive Network (PLNLIN) and 2) Multi-Scale Cross-Content Deeply Discriminative Learning (MCDDL). The PLNLIN is a U-shaped encoder-decoder network, where the proposed new Progressive Local and Non-Local Interactive Module (PLNLIM) is the basic unit in the encoder-decoder framework. The PLNLIM fully explores local and non-local learning in convolution and Transformer operation respectively and the local and non-local content are further interactively learned in a progressive manner. The proposed MCDDL not only discriminates the output of the generator but also receives the deep content from the generator to distinguish real and fake features at each side layer of the discriminator in a multi-scale manner. We show that the proposed MCDDL has fast and stable convergence properties that lack in existing discriminative learning manners. Extensive experiments demonstrate that the proposed method outperforms state-of-the-art methods on five public synthetic datasets and one real-world data. The source codes will be made available at https://github.com/supersupercong/PLNLIN-MCDDL.

## CCS Concepts

• **Computing methodologies** → **Computational photography**.

*Wei Wang is the Corresponding author.

## Keywords

Single image deraining, Multi-Scale Cross-Content Deeply Discriminative Learning, Progressive Interactive Networks, Convolution, Transformer

**ACM Reference Format:**
Cong Wang, Liyan Wang, Jie Mu, Chengjin Yu, and Wei Wang. 2024. Progressive Local and Non-Local Interactive Networks with Deeply Discriminative Training for Image Deraining. In *Proceedings of the 32nd ACM International Conference on Multimedia (MM '24), October 28–November 1, 2024, Melbourne, VIC, Australia.* ACM, New York, NY, USA, 10 pages. https://doi.org/10.1145/3664647.3681399

## 1 Introduction

Image deraining aims to restore a clean image from a given rainy image, which is a typical and challenging image processing problem. Most existing image deraining algorithms depend on the rainy physical model, where the rainy image $O$ can be modeled as the linear combination between the clean background image $B$ and the rain streaks component $R$:

$$O = B + R. \tag{1}$$

Image deraining is a highly ill-posed problem as there exist numerous $B$ and $R$ pairs for a given rainy image $O$. To make this problem well-posed, some priors about rain streaks and rain-free images are proposed [3, 30, 34]. Although these prior-based approaches are effective to some extent, they will fail to work when these priors do not hold on as the priors are usually based on empirical statistical properties of clear images or rain streaks, which do not model inherent properties of the latent clear images.

Current image deraining algorithms are mainly built with deep convolutional neural networks. Although these methods have shown promising results, they still have the following limitations. On the one hand, most existing approaches [7, 29, 36, 50] are designed to learn deep features in a local window by a CNN, which is less effective for modeling non-local information that is important for image deraining [1]. Although some methods [1] model the non-local information by designing Transformers, the single non-local window may not be useful enough as too long-range features may be not relevant enough for the local pixel. On the other hand, generating realistic derained images is still challenging. Although Generative

**Table 1: Summary of different discriminative manners and their properties. ✔ and ✗ respectively denote that the method has and does not have the component. Final and Side respectively denote that the adversarial loss is defined on the final layer and each side layer in the discriminator. Cross-content means that the hierarchical features in the generator are transmitted to the discriminator.**

| Methods | Discriminative Manners | | | Convergence Properties | |
|---------|-------|------|---------------|------|--------|
| | Final | Side | Cross-content | Fast | Stable |
| [35, 69] | ✔ | ✗ | ✗ | ✗ | ✗ |
| [71] | ✔ | ✔ | ✗ | ✗ | ✗ |
| Ours | ✔ | ✔ | ✔ | ✔ | ✔ |

Adversarial Networks (GAN) [9] can provide an effective way for realistic image restoration [35, 69, 70], most existing GAN-based models, e.g., [23, 27, 69], only discriminate the final output of the discriminator to form the adversarial loss, which is a global discriminative manner while neglecting local details discrimination [71]. Moreover, existing GAN-based models usually take the output of the generator as the input of the discriminator, which does not fully explore useful information about the intermediate features of the generator. As the intermediate features from the generator are mainly used for image reconstruction, it is of great interest to develop an algorithm to better explore useful information of the intermediate features to facilitate the estimations of the discriminator for realistic image restoration.

In this paper, we propose a progressive local and non-local interactive deraining network (PLNLIN) with multi-scale cross-content deeply discriminative learning (MCDDL) to solve the image deraining problem. The PLNLIDN is used to learn local and non-local patterns in a progressive interactive manner. Specifically, the PLNLIN is a U-shaped encoder-decoder network with multi-scale hierarchical supervision, where the basic unit in the network is the newly proposed progressive local and Non-Local interactive module (PLNLIM). The PLNLIM fully exploits the convolution and Transformer to respectively learn local and non-local information and the local and non-local content are further interactively learned in a progressive manner. In addition, we introduce multi-scale hierarchical supervision that enforces the multi-scale encoder and decoder features to close with ground-truth, which can make the network more compact for better rain streak removal.

MCDDL is used to overcome unstable training problems and generate better deraining results. The MCDDL not only inputs the output of the generator to the discriminator but also transmits the hierarchical features to the layer in the discriminator. Such a design can make the discriminator deeply discriminate between real and fake features in the generator. Moreover, multi-scale supervised discrimination is also utilized, where adversarial losses are built at each side layer in the discriminator to better help rain streak removal. By designing the MCDDL, the proposed new discriminator has better rain-free image generation ability and fast and stable convergence properties. Table 1 provides the summary of different discriminative manners and their properties.

We summarise the main contributions of this paper as follows:

- We propose a progressive local and non-local interactive network that fully exploits the advantages of convolution and Transformer to solve the image deraining problem.

- We propose a multi-scale cross-content deeply discriminative learning that can discriminate not only the output of the generator but also the deep features of the generator in a multi-scale manner for realistic rain-free image generation.
- We both quantitatively and qualitatively evaluate the proposed method and show that the proposed method outperforms several state-of-the-art methods on five synthetic datasets and one real-world dataset.

## 2 Related Work

ctionLearning-Based Image Deraining With the excellent learning representation ability of CNN in computer vision [14–20, 49, 49, 57–59, 64, 64], CNNs have dominated recent deraining research and achieved great success. Fu et al. [7] first introduce a CNN-based method for image draining. They first use a guided filter to extract high-frequency details and then remove rain content by an end-to-end trainable deep network with residual learning. After that, a series of deep convolutional neural networks are developed [7, 12, 29, 36, 41, 42, 44–48, 61, 62, 72]. The transformer is first introduced by Vaswani et al. [37] for natural language processing. After that, various Transformers are designed for a series of vision tasks, such as image recognition [6], segmentation [56], detection [33, 51], and also image restoration [1, 11, 31, 38, 39, 53, 66].

## 2.1 GAN-Based Image Restoration

Previous research has been demonstrated that GAN [9] can help generate realistic details for image restoration tasks, such as image deblurring [23, 24], low-light image enhancement [13], image super-resolution [25, 70], image dehazing [5, 27, 67], and also image deraining [69, 71]. Different from the above works that only discriminate the final output in discriminator, Zhu et al. [71] explore the multi-scale deeply supervised discriminative manner by imposing constraints at each side layer output of the discriminator to generate the multiple adversarial losses to better discriminate local details and global appearance. Although these GAN-based techniques are able to improve realistic detail restoration to some extent, they ignore the use of intermediate features of the generator, which are vital for realistic image generation. Karnewar and Wang [21] introduces a multi-scale gradient-based GAN for image synthesis by inputting multi-scale images generated by intermediate layers to a single discriminator. We find that this manner fails to work for image restoration as the transferred images are not accurate, limiting to generate high-quality results.

Different from existing GAN-based models, we propose a multi-scale cross-content deeply discriminative learning manner, where we transmit the content features from the generator to the discriminator and discriminate at each side layer in a multi-scale manner.

## 3 Methodology

### 3.1 Overall Framework

Figure 1 shows the proposed progressive local and non-local interactive network with multi-scale cross-content deeply discriminative learning, which is a GAN-based framework. The generator $\mathcal{G}$ is a U-shaped encoder-decoder framework with multi-scale hierarchical supervision. Each block in the encoder and decoder stage is our proposed progressive local and non-local interactive module

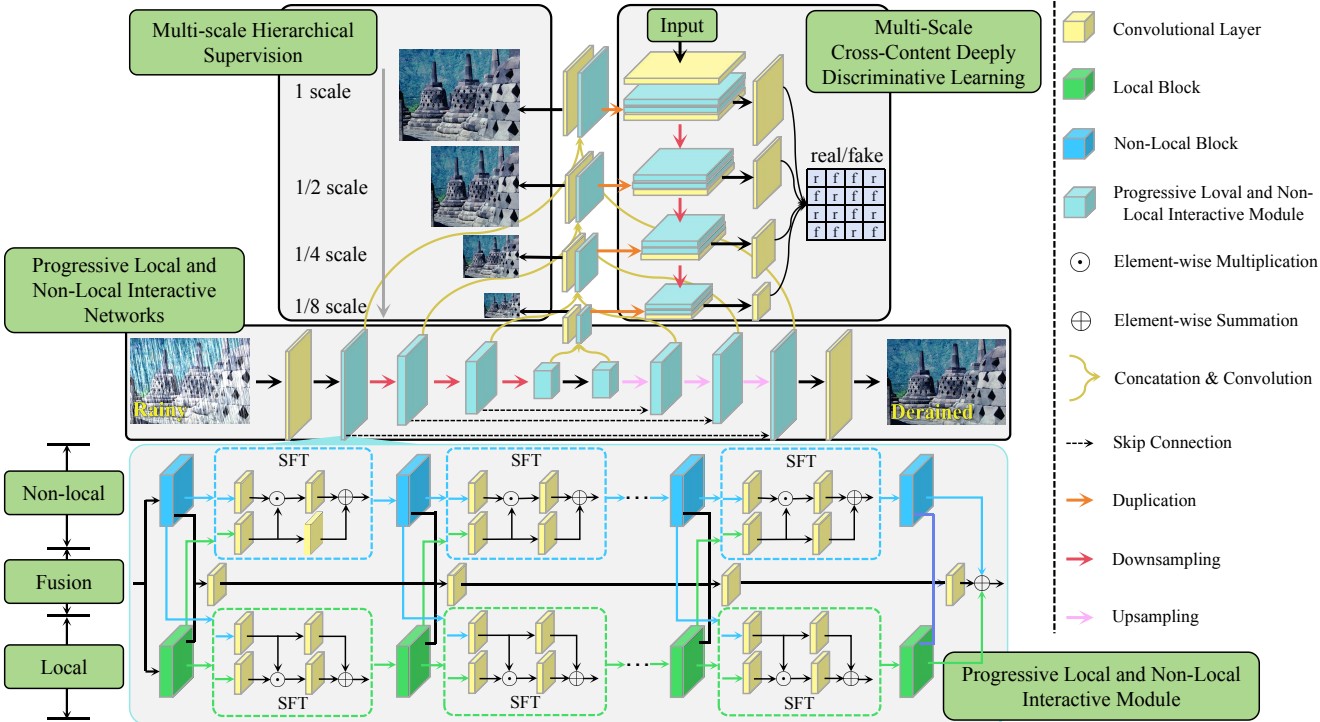

**Figure 1: Overall of Progressive Local and Non-Local Interactive Networks with Multi-Scale Cross-Content Deeply Discriminative Learning, which is a GAN-based framework that contains a generator $\mathcal{G}$ and a discriminator $\mathcal{D}$. $\mathcal{G}$ is a U-shaped encoder-decoder network with multi-scale hierarchical supervision. The multi-scale hierarchical supervision makes the intermediate layers to be more compact for better rain streak removal. $\mathcal{D}$ receives not only the output of the generator but also the hierarchical features from the generator and discriminates on each side layer so that it can help the generator generate more natural features for better rain-free image generation. The basic unit in $\mathcal{G}$ and $\mathcal{D}$ is the proposed Progressive Local and Non-Local Interactive Module that fully explores the local and non-local information and helps better learning for each other.**

(PLNLIM). The PLNLIM adequately explores the advantage of local and non-local operations achieved by residual block [10] and Swin Transformer block [33], respectively. The multi-scale hierarchical supervision is utilized in the generator so that the fusing features at the same level in the encoder and decoder stage are supervised in a multi-scale manner to make the network more compact for better rain streak removal.

The discriminator $\mathcal{D}$ is used to generate realistic results [69], where we propose a new discriminative learning manner that not only takes the output of the generator as the input of the discriminator but also transmits the intermediate features from the generator to the discriminator.

### 3.2 Progressive Local and Non-Local Interactive Module

The Progressive Local and Non-Local Interactive Module (PLNLIM) adequately exploits the advantages of convolution and Transformer to respectively learn the local and non-local information that is further interactively learned in a progressive manner. The PLNLIM has three branches: local branch, non-local branch, and fusion branch. The local branch uses a local window in the residual block

and receives the transferred non-local features from the non-local branch to enhance local features, while the non-local branch learns non-local features and interacts with the features from the local branch to enrich the non-local features. The fusion branch is to fuse local and non-local features to learn useful features. Next, we will introduce the PLNLIM.

We denote $X^0$ as the input feature in the PLNLIM. We first utilize residual block (RB) [10] and Swin Transformer block (STB) [33] to respectively learn the local and non-local information:

$$
\begin{aligned}
X^1_{local} &= \text{RB}(X^0), \\
X^1_{non-local} &= \text{STB}(X^0),
\end{aligned}
\tag{2}
$$

where $X^1_{local}$ and $X^1_{non-local}$ respectively denote the output of RB and STB at the $1^{st}$ stage. Then, the learned local and non-local features are cascaded and fused by $1 \times 1$ convolution in the fusion branch:

$$
X^1_{fusion} = \text{Conv}_{1\times1}\Big(\text{Cancat}[X^1_{local}, X^1_{non-local}]\Big),
\tag{3}
$$

where $\text{Conv}_{1\times1}$ denotes the $1\times1$ convolution and Cancat means the concatenation operation. $X^1_{fusion}$ denotes the output of the fusion branch at the $1^{st}$ stage.

Next, to help local features better utilize non-local information and non-local features better learn useful content from local information, we use Spatial Feature Transform (SFT) layer [52] that can generate affine transformation parameters for spatial-wise feature modulation to interactively boost the learning of local and non-local branch respectively:

$$
\begin{aligned}
Interact^1_{local\&non-local} &= \text{SFT}(X^1_{local}, X^1_{non-local}), \\
Interact^1_{non-local\&local} &= \text{SFT}(X^1_{non-local}, X^1_{local}),
\end{aligned}
\tag{4}
$$

where $Interact^1_{local\&non-local}$ and $Interact^1_{non-local\&local}$ respectively denote the output of local branch interacted with non-local branch and non-local branch interacted with local branch at the $1^{st}$ stage.

Then, the local and non-local contents are interactively learned and the fusion branch is further learned, and all of them are learned in a progressive manner:

$$
\begin{aligned}
X^i_{local} &= \text{SFT}\Big(\text{RB}(X^{i-1}_{local}, X^{i-1}_{non-local})\Big), \\
X^i_{non-local} &= \text{SFT}\Big(\text{STB}(X^{i-1}_{non-local}, X^{i-1}_{local})\Big), \\
X^i_{fusion} &= \text{Conv}_{1\times1}\Big(\text{Cancat}[X^{i-1}_{fusion}, X^i_{local}, X^i_{non-local}]\Big),
\end{aligned}
\tag{5}
$$

where $X^i_{local}$, $X^i_{non-local}$, and $X^i_{fusion}$ $(i = 2, \cdots, N)$ respectively denote the output of the local branch, the non-local branch, and the fusion branch at the $i^{th}$ progressive stage.

Finally, the learned features of these three branches are added by respective learning manner at the final stage:

$$
\begin{aligned}
X^{\text{final}} = \quad & \text{RB}(X^N_{local}) + \text{STB}(X^N_{non-local}) + \\
& \text{Conv}_{1\times1}\Big(\text{Cancat}[X^N_{fusion}, X^N_{local}, X^N_{non-local}]\Big),
\end{aligned}
\tag{6}
$$

where $X^{\text{final}}$ is the output of the PLNLIM.

By such a design, the local window not only captures local patterns but also interacts with non-local information to learn useful features in the long-range content meanwhile the non-local window can not only perceive global information but also receive local content from the local window. The local and non-local features are interactively learned, which can remedy the shortage of each other for better rain streak removal.

## 3.3 Multi-Scale Cross-Content Deeply Discriminative Learning

Different from existing discriminators [9, 69, 71] that only take the output of the generator as the input of the discriminator, we develop a new multi-scale cross-content deeply discriminative learning (MCDDL) by transmitting the content of intermediate layers in the generator to the discriminator and discriminating on each side layer in the discriminator.

With the hierarchical contents from the generator, the MCDDL fuses them to help the discriminator better discriminate between real and fake content from the generator, which can be expressed as:

$$
\begin{aligned}
\tilde{F}^j_D &= \text{PLNLIM}\Big(\text{Concat}[F^j_H, F^j_D]\Big), \\
F^{j+1}_D &= \text{Pooling}(\tilde{F}^j_D), \ j = 0, 1, 2, 3, 4,
\end{aligned}
\tag{7}
$$

where $\tilde{F}^j_D$ denotes the output features at $j^{th}$ layer in the discriminator; $F^{j+1}_D$ denotes the input features of $(j + 1)^{th}$ layer that is

**Algorithm 1** Training Process of the Proposed PLNLIN-MCDDL

---

**Preparation:** Rainy images $O$ and corresponding rain-free ground-truth images $B$

**Input:** $O$

**Output:** Derained images $\hat{B}$

1: **While** epoch $\leq$ epoch$_{\text{max}}$ **do**:
2:     Obtain fake label images and features: $\mathbf{Y} = \mathcal{G}(O)$; Obtain real label images and features: $\mathbf{Z} = \mathcal{G}(B)$,
3:     **if** epoch % $r$ == 0:
4:         Update $\mathcal{G}$ via (9)
5:         Update $\mathcal{D}$ via (12)
6:     **else**:
7:         Update $\mathcal{G}$ via (9)
8:     epoch $\leftarrow$ epoch +1
9: **End while**

Output derained images: $\hat{B}$

---

pooled by $j^{th}$ layer in the discriminator; $F^j_H$ denotes the hierarchical features at $j^{th}$ layer in generator. Similar to [69], we use Pooling operation with the size of $2 \times 2$ stride and $2 \times 2$ kernel between two adjacent layers. Here, we have 4 layers in the discriminator. When $j = 0$, (7) is defined as: $F^1_D = \text{PLNLIM}(Image)$ that takes the image as the input of the discriminator. When $j = 4$, (7) is defined as: $\tilde{F}^4_D = \text{PLNLIM}\Big(\text{Concat}[F^4_H, F^4_D]\Big)$.

Note that (7) is different from existing discriminators [69, 71] that only take the output image of the generator as the input of the discriminators without considering the feature content from the generator, which can be expressed as: $\tilde{F}^j_D = \text{PLNLIM}(F^j_D)$ and $F^{j+1}_D = \text{Pooling}(\tilde{F}^j_D)$, $(j = 0, 1, 2, 3, 4)$. Hence, our discriminator achieves cross-content discrimination by transmitting the content from the generator to the discriminator, which can help the discriminator deeply discriminate between real and fake features so that the generator can generate more natural features for better rain-free image generation.

We utilize multi-scale supervised adversarial losses to discriminate the features at each side layer:

$$
\begin{aligned}
\mathcal{L}^l_{\text{adversarial}} = \quad & -\mathbb{E}_{\mathbf{Z}}\Big[\log\big(1 - D_{ra}(\mathbf{Z}^l; \mathbf{Y}^l)\big)\Big] - \\
& \mathbb{E}_{\mathbf{Y}}\Big[\log\big(D_{ra}(\mathbf{Y}^l; \mathbf{Z}^l)\big)\Big], \ l = 1, 2, 3, 4,
\end{aligned}
\tag{8}
$$

where $\mathbf{Y}$ denotes the fake label that is the joint of derained images and reconstructed features, while $\mathbf{Z}$ is the corresponding real label that is the joint of rain-free images and rain-free features. $\mathbf{Y}^l$ and $\mathbf{Z}^l$ respectively denote the fake and real features at $l^{th}$ side layer in the discriminator. $D_{ra}(\mathbf{P}; \mathbf{Q}) = \text{sigmoid}\Big(\mathcal{D}(\mathbf{P}) - \mathbb{E}_{\mathbf{Q}}\big[\mathcal{D}(\mathbf{Q})\big]\Big)$. $\mathcal{D}(\cdot)$ is the discriminator. However, we can only obtain real-label images with no means to acquire rain-free features. To this end, we generate rain-free features as real features by inputting the rain-free image to the generator.

**Table 2: Quantitative results on five synthetic datasets. ↑ denotes higher is better. The best results are marked in bold.**

| Methods | Rain200H | | Rain200L | | Rain1200 | | Rain1400 | | Rain12 | | # Parameters |
|---|---|---|---|---|---|---|---|---|---|---|---|
| | PSNR ↑ | SSIM ↑ | PSNR ↑ | SSIM ↑ | PSNR ↑ | SSIM ↑ | PSNR ↑ | SSIM ↑ | PSNR ↑ | SSIM ↑ | |
| RESCAN [29] | 26.661 | 0.8419 | 36.993 | 0.9788 | 32.127 | 0.9028 | 30.969 | 0.9117 | 32.965 | 0.9545 | 0.15M |
| NLEDN [26] | 27.315 | 0.8904 | 36.487 | 0.9792 | 32.473 | 0.9198 | 31.014 | 0.9206 | 33.028 | 0.9615 | 1.01M |
| SSIR [54] | 14.420 | 0.4501 | 23.476 | 0.8026 | 24.427 | 0.7713 | 25.772 | 0.8224 | 24.138 | 0.7768 | 0.06M |
| PreNet [36] | 27.525 | 0.8663 | 34.266 | 0.9660 | 30.456 | 0.8702 | 30.984 | 0.9156 | 35.095 | 0.9400 | 0.17M |
| SpaNet [50] | 25.484 | 0.8584 | 36.075 | 0.9774 | 27.099 | 0.8082 | 29.000 | 0.8891 | 33.217 | 0.9546 | 0.28M |
| DCSFN [44] | 28.469 | 0.9016 | 37.847 | 0.9842 | 32.275 | 0.9228 | 31.493 | 0.9279 | 35.803 | 0.9683 | 6.45M |
| MSPFN [12] | 25.553 | 0.8039 | 30.367 | 0.9219 | 30.382 | 0.8860 | 31.514 | 0.9203 | 34.253 | 0.9469 | 21.00M |
| RCDNet [48] | 28.698 | 0.8904 | 38.400 | 0.9841 | 32.273 | 0.9111 | 31.016 | 0.9164 | 31.038 | 0.9069 | 3.67M |
| Syn2Real [63] | 14.495 | 0.4021 | 31.035 | 0.9365 | 28.812 | 0.8400 | 28.582 | 0.8586 | 28.434 | 0.9038 | 2.62M |
| MPRNet [65] | 29.949 | 0.9151 | 36.610 | 0.9785 | 33.655 | 0.9310 | 32.257 | 0.9325 | 36.578 | 0.9696 | 3.64M |
| DualGCN [8] | 28.758 | 0.9026 | 38.415 | 0.9818 | 32.033 | 0.9163 | 30.567 | 0.9148 | 35.805 | 0.9687 | 2.73M |
| SwinIR [32] | 29.574 | 0.9049 | 39.282 | 0.9869 | 32.974 | 0.9298 | 31.997 | 0.9304 | 36.362 | 0.9698 | 11.5M |
| SSID-KD [4] | 28.706 | 0.9005 | 38.778 | 0.9864 | 32.424 | 0.9202 | 30.540 | 0.9136 | 35.473 | 0.9682 | 4.43M |
| RadNet [55] | 30.080 | 0.9150 | 38.680 | 0.9850 | - | - | - | - | - | - | - |
| Wang et al. [40] | 29.985 | 0.9218 | 39.284 | 0.9875 | 33.718 | 0.9327 | 32.617 | 0.9334 | 36.851 | 0.9714 | 2.04M |
| HCT-FFN [2] | 28.987 | 0.8933 | 37.426 | 0.9798 | - | - | - | - | 34.570 | 0.9453 | 0.87M |
| **Ours** | **30.164** | **0.9227** | **39.289** | **0.9893** | **33.731** | **0.9338** | **32.643** | **0.9341** | **37.211** | **0.9734** | 2.25M |

## 3.4 Loss Function

To train the network, we utilize the following objective loss functions including image pixel reconstruction loss, multi-scale hierarchical supervision loss, and adversarial loss:

$$\mathcal{L} = \mathcal{L}_{\text{pixel}} + \alpha \mathcal{L}_{\text{hierarchical}} + \beta \mathcal{L}_{\text{adversarial}}, \tag{9}$$

where $\alpha$ and $\beta$ denote the hyper-parameters. In the following, we explain each term in detail.

**Image pixel reconstruction loss** $\mathcal{L}_{\text{pixel}}$. Followed [36, 44] that SSIM-based loss has achieved better performance, we use it as the image pixel reconstruction loss:

$$\mathcal{L}_{\text{pixel}} = 1 - \text{SSIM}(\hat{B}, B), \tag{10}$$

where $\hat{B}$ and $B$ denote the estimated derained image and corresponding ground-truth.

**Multi-scale hierarchical supervision loss** $\mathcal{L}_{\text{hierarchical}}$. As the image pixel reconstruction loss only constrains the network output that may not generate satisfactory results, we propose a multi-scale hierarchical supervision loss on the intermediate layers to enforce the network more compact for better rain streak removal:

$$\mathcal{L}_{\text{hierarchical}} = \sum_{k=1}^{4} \lambda_k \left( 1 - \text{SSIM}(\hat{B}_{\frac{1}{2^{k-1}}}, B_{\frac{1}{2^{k-1}}}) \right), \tag{11}$$

where $\hat{B}_{\frac{1}{2^{k-1}}} = \text{Conv}(\text{PLNLIM}(\text{Concat}(E_k, D_k)))$ is the $\frac{1}{2^{k-1}}$ scale output, while $B_{\frac{1}{2^{k-1}}}$ is the corresponding ground-truth. Conv denotes a 3×3 convolution. $E_k$ and $D_k$ respectively refer to the feature at $k^{th}$ stage of encoder and decoder. $\lambda_1, \lambda_2, \lambda_3,$ and $\lambda_4$ are set as 0.1, 0.05, 0.02, and 0.01, respectively.

**Multi-scale cross-content deeply discriminative adversarial losses** $\mathcal{L}_{\text{adversarial}}$. As defined each adversarial loss at each side layer in the discriminator in (8), the total adversarial loss is:

$$\mathcal{L}_{\text{adversarial}} = \sum_{l=1}^{4} \mu_l \mathcal{L}_{\text{adversarial}}^l, \tag{12}$$

where $\mu_1, \mu_2, \mu_3,$ and $\mu_4$ are respectively set as 0.0005, 0.001, 0.0025, and 0.005.

## 4 Experiments

In this section, we provide more experiments to demonstrate the effectiveness of our proposed approach.

## 4.1 Implementation Details

We set the number of channels as 32 in each layer except the last one in both generator and discriminator. We respectively set the number of the channel of the last layer as 3 in the generator and 1 in the discriminator. The radio $r$ of updating the generator and discriminator is 5. In the encoder stage of the generator, $N$ in the PLNLIM is respectively set as 8, 4, 2, 2, and the decoder stage has a symmetric structure with the encoder. In the discriminator, $N$ is set as 2 for all PLNLIMs. We randomly crop an image patch of the size 112×112 pixels. The batch size is 2. We use the ADAM [22] optimizer with default parameters to train the proposed network. The initial learning rate is 0.0001, which is respectively divided by 10 at 300 and 400 epochs, and the model training terminates after 500 epochs, i.e., $\text{epoch}_{\max} = 500$. The values of $\alpha$ and $\beta$ are empirically set to be 1. The training process is illustrated in Algorithm 1.

## 4.2 Datasets

**Synthetic Dataset.** We adopt five widely used synthetic datasets: Rain200H [60], Rain200L [60], Rain1200 [68], Rain1400 [7], and Rain12 [30] to evaluate the deraining performance. As Rain12 only contains 12 testing samples, we use the models trained on Rain200H to evaluate the restoration quality of this dataset.

**Real-world Data.** [28, 43, 60] collect a mass of real-world rainy images from the Internet. We use them as the real-world dataset.

## 4.3 Results on Synthetic Datasets

Table 2 summarises the deraining performance on the five synthetic datasets, where our method performs better than state-of-the-art methods in terms of PSNR and SSIM. We also provide several visual examples on the most challenging Rain200H dataset in Figure 2. Note that our method is able to generate clearer results with better texture and details, while other state-of-the-art approaches generate

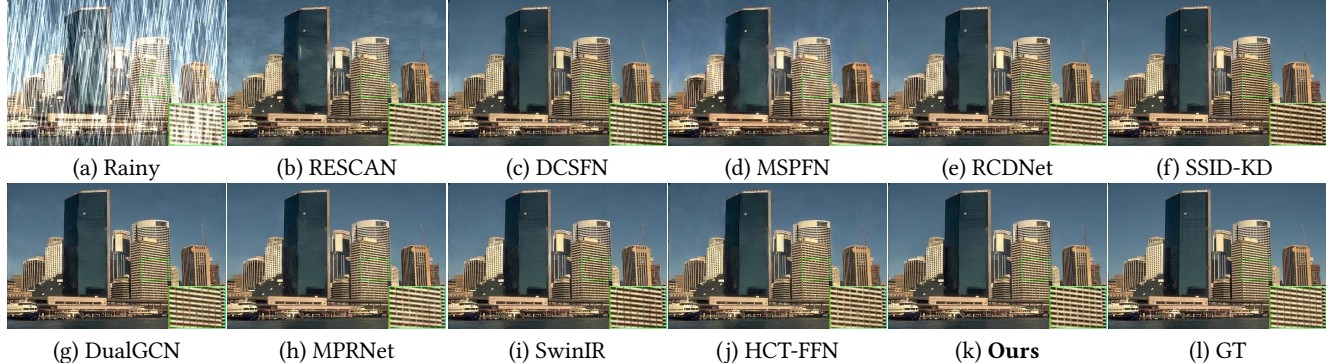

(a) Rainy          (b) RESCAN          (c) DCSFN          (d) MSPFN          (e) RCDNet          (f) SSID-KD

(g) DualGCN          (h) MPRNet          (i) SwinIR          (j) HCT-FFN          (k) **Ours**          (l) GT

**Figure 2: Results on the most challenging dataset Rain200H [60]. The proposed algorithm is able to preserve better structures.**

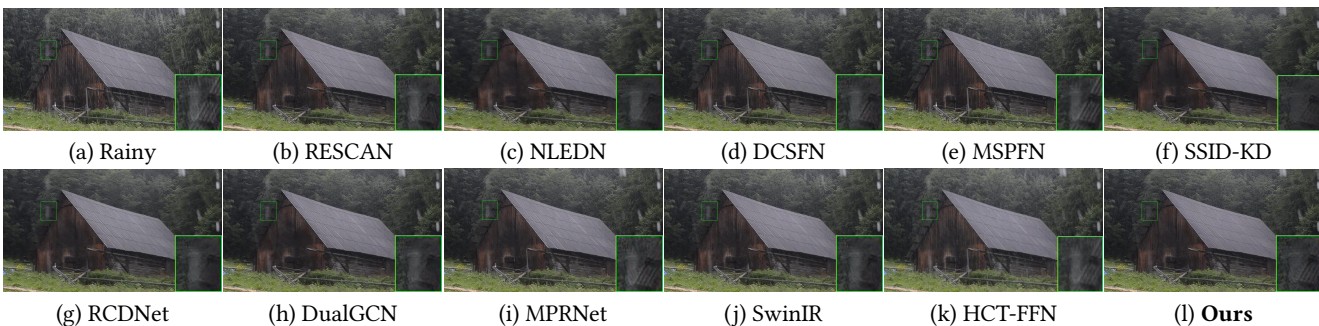

(a) Rainy          (b) RESCAN          (c) NLEDN          (d) DCSFN          (e) MSPFN          (f) SSID-KD

(g) RCDNet          (h) DualGCN          (i) MPRNet          (j) SwinIR          (k) HCT-FFN          (l) **Ours**

**Figure 3: Results on one real-world data. The proposed algorithm is able to generate results with fewer artifacts.**

the results with some artifacts. These quantitative and qualitative results demonstrate the effectiveness of the proposed method.

## 4.4 Results on Real-World Data

To examine the generalization of the proposed algorithm in real-world scenarios, we present two real-world rainy examples in Figure 3. We note that the proposed method is able to generate cleaner deraining results than state-of-the-art methods, especially in the marked regions.

## 4.5 Ablation Study

This section provides ablation studies about the proposed method.

*4.5.1 Analysis on Progressive Local and Non-Local Interactive Module.* As the proposed progressive Local and Non-Local interactive module (PLNLIM) contains the local branch, the non-local branch, the fusion branch, and the interactive learning operation between local and non-local branches (i.e., the SFT layer), we need to examine the effect of these components by disabling them in the proposed method. For fair comparisons, different models have an almost equal number of parameters by adjusting the number of channels. Table 3 shows the evaluation results. The comparisons between the baselines show that our module (i.e., (f)) has a better performance compared with the module without the fusion branch (i.e., (a)). The results get worse when we use the concatenation operation (i.e., (b)) to replace the SFT layer or remove the SFT layer (i.e., (e) means that the local branch and the non-local branch are

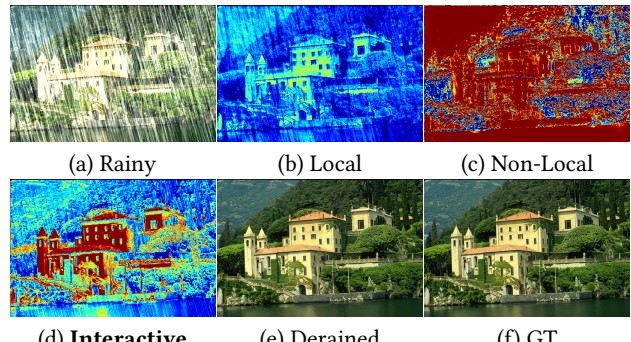

(a) Rainy          (b) Local          (c) Non-Local

(d) **Interactive**          (e) Derained          (f) GT

**Figure 4: Feature visualization on the effect of the progressive local and non-local interactive module.**

independent). Furthermore, we observe that the interactive learning between local and non-local branches is able to help improve deraining quality compared with the models of using both single local learning (i.e., (c) that is to replace Swin Transformer blocks as residual blocks) or both single non-local learning (i.e., (d)) that is to replace residual blocks as Swin Transformer blocks) in the PLNLIM. This demonstrates that the proposed interactive learning between local and non-local information plays an important role in image deraining. The above results show that each component we used is beneficial to image deraining.

We further show some intermediate features to demonstrate the effect of the PLNLIM. Figure 4 shows that the local operation

**Table 3: Ablation results on our progressive local and non-local interactive module.**

| Experiments | (a) | (b) | (c) | (d) | (e) | (f) (Ours) |
|---|---|---|---|---|---|---|
| Fusion branch | ✗ | ✗ | ✓ | ✓ | ✓ | ✓ |
| SFT → Concatenation | ✗ | ✓ | ✗ | ✗ | ✗ | ✗ |
| SFT layer | ✓ | ✗ | ✓ | ✓ | ✗ | ✓ |
| Both local learning, i.e, STB → RB | ✗ | ✗ | ✓ | ✗ | ✗ | ✗ |
| Both non-local learning, i.e, RB → STB | ✗ | ✗ | ✗ | ✓ | ✗ | ✗ |
| Independent learning in local and non-local branches, i.e., w/o SFT layer | ✗ | ✗ | ✗ | ✗ | ✓ | ✗ |
| Interactive learning between local and non-local branches | ✓ | ✓ | ✗ | ✗ | ✗ | ✓ |
| PSNR ↑ | 30.051 | 30.012 | 29.360 | 29.553 | 29.957 | 30.164 |
| SSIM ↑ | 0.9212 | 0.9208 | 0.9137 | 0.9142 | 0.9169 | 0.9227 |

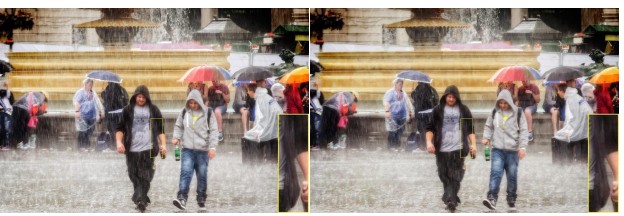

(a) Rainy                    (b) Both local learning

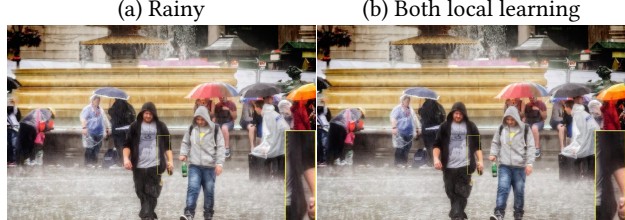

(c) Both non-local learning        (d) **Interactive learning**

**Figure 5: A real-world comparison example of both local learning, both non-local learning, and interactive learning.**

**Table 4: Effectiveness on progressive local and non-local interactive module.**

| Model | Local | Non-Local | **Ours** |
|---|---|---|---|
| PSNR ↑ | 29.866 | 29.948 | **30.164** |
| SSIM ↑ | 0.9190 | 0.9194 | **0.9227** |

**Table 5: Effect on the multi-scale hierarchical supervision.**

| Model | w/o (11) | w/ (11) |
|---|---|---|
| PSNR ↑ | 29.922 | **30.164** |
| SSIM ↑ | 0.9192 | **0.9227** |

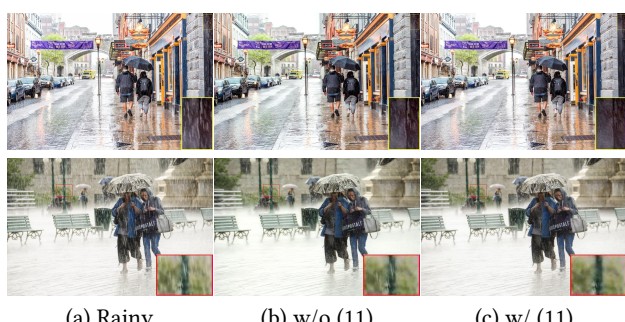

(a) Rainy          (b) w/o (11)          (c) w/ (11)

**Figure 6: Two real-world examples on the effect of multi-scale hierarchical supervision.**

cannot capture global attention information well (i.e., Figure 4(b)). Although the non-local operation can capture global attention information, it tends to lose some local structures (i.e., Figure 4(c)). The interactive learning between the local and non-local operations is able to produce better structures and contents (i.e., Figure 4(d)), which will help produce a better deraining result.

Figure 5 shows the effect of single local learning, single non-local learning, and interactive learning on a real-world example. Note that both single local learning (i.e., Figure 5(b)) and single non-local learning (i.e., Figure 5(c)) always hand down some rain streaks, while the proposed interactive learning between local and non-local branches is able to produce a cleaner result (Figure 5(d)). This example also demonstrates that the proposed interactive learning plays a positive role in rain-free image generation in real-world conditions.

To further examine the effectiveness of the PLNLIM, we use residual blocks [10] or Swin Transformer blocks [33] to replace the PLNLIMs in the generator. Note that we also adjust the number of channels so that different models have an almost equal number of parameters for fair comparisons. Table 4 shows that our PLNLIM is a better module, while both residual blocks or Swin Transformer blocks are not as good as ours. This demonstrates that both single convolution operation and single Transformer architecture cannot effectively generate high-quality rain-free images, while the interactive learning between convolution and Transformer is a powerful tool for improving deraining performance.

*4.5.2 Analysis on Different Discriminative Manners.* As we propose a new discriminative manner with multi-scale cross-content deeply discriminative learning, we need to compare it with existing manners [9, 71]. Furthermore, we also provide the adversarial loss curves of the generator in Figure 7. It is worth noting that our proposed discriminative manner has more stable and faster convergence properties on each side layer that are lacking in other discriminative manners. This also shows that our proposed discriminative manner can effectively solve the unstable training process problem that exists in most existing discriminative manners. Figure 8 shows two real-world examples of different discriminative manners, where the proposed discriminative manner is able to help produce cleaner results. These results and examples demonstrate the effectiveness of our proposed discriminative manner that not only has fast and stable convergence properties but also improves

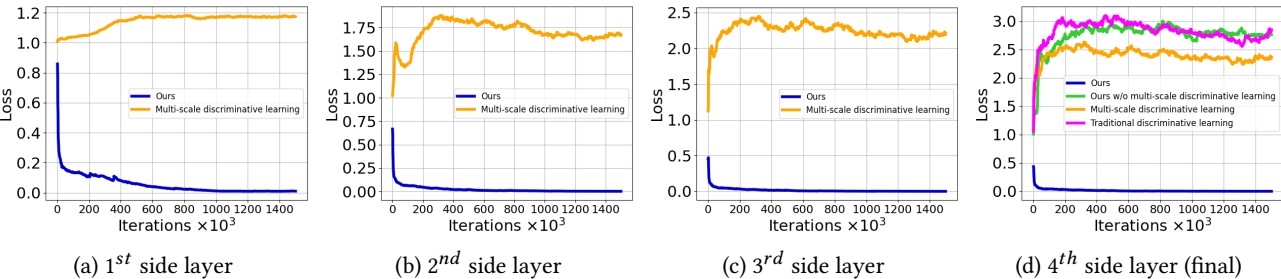

(a) $1^{st}$ side layer      (b) $2^{nd}$ side layer      (c) $3^{rd}$ side layer      (d) $4^{th}$ side layer (final)

**Figure 7: Loss curves for different adversarial losses of the generator. $i^{th}$ side layer denotes the $i^{th}$ side layer in the discriminator.**

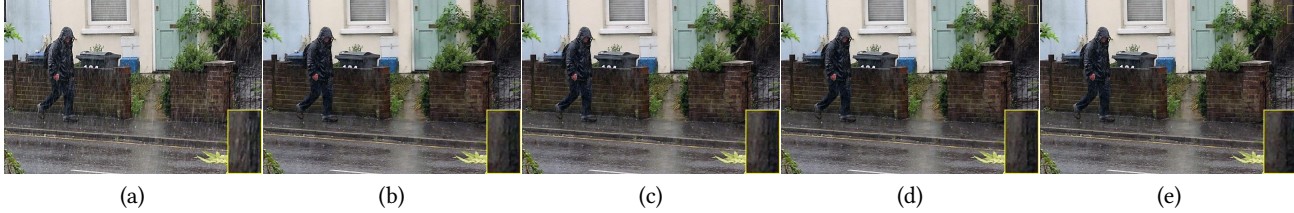

(a)      (b)      (c)      (d)      (e)

**Figure 8: One real-world example of different discriminative manners. (a)-(e) are respectively the rainy input, the results of traditional discriminative learning [9], multi-scale deeply discriminative learning [71], ours w/o multi-scale deeply discriminative learning, and our proposed final discriminative learning.**

**Table 6: Effect on progressive local and non-local interactive module in the discriminator.**

| Model | Local | Non-local | **Ours** |
|---|---|---|---|
| PSNR ↑ | 29.983 | 29.992 | **30.164** |
| SSIM ↑ | 0.9174 | 0.9175 | **0.9227** |

**Table 7: Effect on the updated radio $r$ between the training generator and discriminator.**

| $r$ | 1 | 2 | 3 | **5** | 10 |
|---|---|---|---|---|---|
| PSNR ↑ | 29.824 | 29.932 | 30.012 | **30.164** | 30.004 |
| SSIM ↑ | 0.9098 | 0.9102 | 0.9193 | **0.9227** | 0.9187 |

the deraining quality. We also hope that the proposed discriminative manner can help other vision problems for better vision learning.

*4.5.3 Effect on Multi-Scale Hierarchical Supervision.* One may wonder what is the effect of the multi-scale hierarchical supervision (i.e., (11)) and which effect it produces if we remove the supervision. Table 5 answers this question, where we find that multi-scale hierarchical supervision has a positive effect on rain streak removal. We also provide several real-world examples of the effect of the multi-scale hierarchical supervision in Figure 6. Observe that the multi-scale hierarchical supervision can help recover cleaner results in real-world conditions. These results demonstrate that the introduced multi-scale hierarchical supervision is meaningful for image deraining on both synthetic and real-world images.

*4.5.4 Effect on Progressive Local and Non-Local Interactive Module in the Discriminator.* As we also use PLNLIM in the discriminator, we need to verify its effect compared with convolution-based and Transformer-based discriminators. Table 6 summarises the results. We note that using PLNLIM is better than both convolution and

Transformer in the discriminator for rain streak removal. Hence, we conclude that the proposed PLNLIM is a better basic unit in neural networks for deraining.

*4.5.5 Effect on the Updated Radio $r$ between the Training Generator and Discriminator.* Table 7 shows the effect on the updated radio $r$ between the training generator and discriminator. We note that the model obtains better performance when $r$ is 5.

## 5 Conclusion

We have proposed a progressive local and non-local interactive deraining network (PLNLIN) with multi-scale cross-content deeply discriminative learning (MCDDL) for image deraining. The PLNLIN fully exploits the advantages of convolution and Transformer operations to respectively learn local and non-local information that is further interactively learned in a progressive manner. The MCDDL that not only receives the features from the generator but also deeply discriminates each side layer in the discriminator has been verified to help rain streak removal and has fast and stable convergence properties that lack in existing discriminative learning manners. Extensive experiments have demonstrated that our algorithm outperforms state-of-the-art methods on both synthetic datasets and real-world data.

## Acknowledgements

We thank the anonymous reviewers for their constructive suggestions. This work was supported by the National Natural Science Foundation of China No. 62306343, Shenzhen Science and Technology Program (No. KQTD20221101093559018), and Fundamental Research Funds for the Central Universities, Sun Yat-sen University under Grants No. 23qnpy56.

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
