# OpenReview forum: "Progressive Local and Non-Local Interactive Networks with Deeply Discriminative Training for Image Deraining"
_acmmm.org/ACMMM/2024/Conference — MM2024 Poster_

### Official Review · Reviewer_T8T7 · 2024-05-13

**Rating:** 6
**Confidence:** 4

**Summary:**

This paper proposes Progressive Local and Non-Local Interactive Networks with Multi-Scale Cross-Content Deeply Discriminative Learning for Image Deraining.
To better discriminate the rainy information in the deep feature space, this paper proposes Multi-Scale Cross-Content Deeply Discriminative Learning, which is different from existing GAN and more effective and more stable.

Overall, the Multi-Scale Cross-Content Deeply Discriminative Learning seems to be more interesting.

**Strengths:**

This paper is well written and the results are SOTA.
The experiments are adequate and efficient.
The Multi-Scale Cross-Content Deeply Discriminative Learning looks interesting and effective and meaningful.

**Limitations:**

The overall architecture is complicated, but the Multi-Scale Cross-Content Deeply Discriminative Learning is interesting. It would be better if the authors can give more detailed explorations about the Multi-Scale Cross-Content Deeply Discriminative Learning.
The authors claim that the introduced MCDDL overcomes the "unstable training problem". Could the authors elaborate on this point?

**Suitability:**

2

---

### Official Review · Reviewer_nYFR · 2024-05-20

**Rating:** 5
**Confidence:** 3

**Summary:**

This paper proposes a progressive local and non-local interactive network (PLNLIN) based on multi-scale cross-content deeply
discriminative learning (MCDDL) for image deraining. The PLNLIN uses convolution and Transformer to respectively get local and non-local features, which further  interactively transferred to SFT. It is rational and demonstrated effective by Table 2. The paper also tries to present a multi-scale cross-content deeply discrimnative learning (MCDDL), which seems not so sound for its poor clarity.

**Strengths:**

This proposal is novel to obtain local and non-local features progressively by convolution and Transformer, then process them interactively by SFT. Local and non-local features help to generate rain-free images with more details.  It is rational and demonstrated effective by Table 2.

**Limitations:**

The paper also tries to present a multi-scale cross-content deeply discrimnative learning (MCDDL), which seems not so sound for its poor clarity. The MCDDL may be not illustrated clearly by Figure 1. What following Multi-scale Hierarchical Supervision?
And there are some issues else,
1 The title is too long.
2 It is inappropriate that the phrase Single image deraining is listed as one keyword but doesn't appear in the abstract.
3 The Keywords are too long. They are supposed to be words rather than phrases, or shorter.
4 In line 195，the reference citation is missing.
5 In section 2.1 the content is not very relevant to the section title.
6 It is suggested to simplify and shorten sentences for comprehension, such as the sentences at lines from 186 to 189, from 189 to 194, from 219 to 223, and so on.
7 Language presentation is not so good, such as
 in line 197 ——We find that this manner fails to work for image restoration as the transferred images
are not accurate, limiting to generate high-quality results. What is the transferred images？ What subject is for verb limit?
in line 20——We will show that such a design is able to improve the image-deraining quality and the discriminator has faster and morestable convergence properties that lack in existing discriminative approaches.
In line 182——Previous research has been demonstrated that GAN......
In line 195——Karnewar and Wang introduces a multi-scale gradient-based GAN......
Please have your submission proof-read for English style and grammar issues, especially for those long sentences. There are also some slips such as ratio/radio.
8 In Table 2，the experiments labels are not accordant to section 4.5.1.

**Suitability:**

3

---

### Official Review · Reviewer_Dkw4 · 2024-05-21

**Rating:** 5
**Confidence:** 4

**Summary:**

This paper proposes a progressive local and non-local interactive network with multi-scale cross-content deeply discriminative learning to solve image deraining. The proposed model contains 1) Progressive Local and Non-Local Interactive Network (PLNLIN) and 2) Multi-Scale Cross-Content Deeply Discriminative Learning (MCDDL). The PLNLIN is a U-shaped encoder-decoder network with the proposed new Progressive Local and Non-Local Interactive Module (PLNLIM) as the basic unit in the encoder-decoder framework. The PLNLIM fully explores local and non-local learning in convolution and Transformer operation respectively and the local and non-local content are further interactively learned in a progressive manner. The proposed MCDDL not only discriminates the output of the generator but also receives the deep content from the generator to distinguish real and fake features at each side layer of the discriminator in a multi-scale manner. The experiments demonstrate that the proposed method outperforms state-of-the-art methods on five public synthetic datasets and one real-world dataset.

**Strengths:**

1, The proposed MCDDL is interesting. According to the shown loss curves, it can solve the unstable problem of GAN.

2, This paper is easy to understand.

3, The experiments are enough and the ablation study demonstrates the effectiveness of the proposed component.

4, The experimental results look better.

**Limitations:**

Although the proposed network looks interesting, I still have some concerns.

1, The proposed MCDDL seems a general technique. Can it be applied to some tasks, such as image dehazing?

2, The proposed Local and Non-Local Interactive Network is not a novel architecture. Hence I suggest that the authors should re-write the title and the contributions.

3, The authors should compare with methods in 2023.

**Suitability:**

2

---

### Meta-Review · Area_Chair_AzkA · 2024-07-02

**Recommendation:** Accept (Poster)
**Confidence:** 5

**Metareview:**

This paper receives all positive rating scores. The paper is strong in its contributions and experimental validation but requires minor revisions for clarity, language, and additional validation to ensure broader applicability. The suggested improvements will enhance the paper's readability and impact, making it a valuable contribution to the field.